# Interpreting the pervasive observation of U-shaped Site Frequency Spectra

**Fabian Freund**[1,2☯], **Elise Kerdoncuff**[3,10☯], **Sebastian Matuszewski**[4], **Marguerite Lapierre**[10], **Marcel Hildebrandt**[5], **Jeffrey D. Jensen**[6], **Luca Ferretti**[7], **Amaury Lambert**[8,10], **Timothy B. Sackton**[9], **Guillaume Achaz**[10,11]*

1 Institute of Plant Breeding, Seed Science and Population Genetics, University of Hohenheim, Stuttgart, Germany, 2 Department of Genetics and Genome Biology, University of Leicester, Leicester, United Kingdom, 3 Department of Genetics, University of California, Berkeley, California, United States of America, 4 Accenture, Vienna, Austria, 5 Siemens AG, Munich, Germany, 6 Center for Evolution & Medicine, School of Life Sciences, Arizona State University, Tempe, Arizona, United States of America, 7 Big Data Institute, Li Ka Shing Centre for Health Information and Discovery, Nuffield Department of Medicine, University of Oxford, Oxford, United Kingdom, 8 Institut de Biologie de l'ENS (IBENS), École Normale Supérieure, Paris, France, 9 Informatics Group, Harvard University, Cambridge, Massachusetts, United States of America, 10 SMILE group, Center for Interdisciplinary Research in Biology (CIRB), Collège de France, Paris, France, 11 Éco-anthropologie, Muséum National d'Histoire Naturelle, Université Paris-Cité, Paris, France

☯ These authors contributed equally to this work.
* guillaume.achaz@mnhn.fr

**Data Availability Statement:** All used genomic data are publicly available. All processed material is provided in the supplementary (e.g. tables and

## Abstract

The standard neutral model of molecular evolution has traditionally been used as the null model for population genomics. We gathered a collection of 45 genome-wide site frequency spectra from a diverse set of species, most of which display an excess of low and high frequency variants compared to the expectation of the standard neutral model, resulting in U-shaped spectra. We show that multiple merger coalescent models often provide a better fit to these observations than the standard Kingman coalescent. Hence, in many circumstances these under-utilized models may serve as the more appropriate reference for genomic analyses. We further discuss the underlying evolutionary processes that may result in the widespread U-shape of frequency spectra.

## Author summary

This study investigates the assumed universality of the standard neutral model of molecular evolution. We demonstrate that genealogical models alternative to the widely used Kingman coalescent often provide greatly improved fits to observed genome-wide allele frequency data for taxa sampled widely from across the tree of life. As such, we argue that these more generalized multiple merger models (which contain the Kingman coalescent as a special case) may prove more fruitful and appropriate in future population genomic studies. Importantly, this modification of the standard model for interpreting genetic diversity has potentially profound implications for many population genetic inference approaches (e.g., scanning for targets of selection

plots) and in our code and data repository https://github.com/fabfreund/usfs_mmc.

**Funding:** We thank NSF for support through the DEB-1754397 to Timothy B. Sackton and DFG for support through FR 3633/2-1 (within Priority Program 1590: Probabilistic Structures in Evolution) to Fabian Freund. Jeffrey D. Jensen was supported by National Institutes of Health grant R35GM139383 The funders had no role in study design, data collection and analysis, decision to publish, or preparation of the manuscript.

**Competing interests:** None.

across the genome and reconstructing population history), as well as for analyses in related fields.

## 1 Introduction

The Kingman coalescent, [1], a stochastic process describing the distribution of random, bifurcating genealogical trees in a Wright-Fisher population, has been enormously impactful in the study of natural genetic variation in populations [2]. Under the standard neutral theory [3, 4], the coalescent can be used to derive expectations of neutral diversity by tracking mutations along the branches of random genealogies, and extensions can accommodate complex processes such as recombination [5], population structure [6], and natural selection [7]. The power of this approach relies on being able to compare deviations observed in real data from expectations under the coalescent model.

One common metric used to study the consistency between the assumptions of this model and the observed data is the *Site Frequency Spectrum* (SFS)—that is, the distribution of mutational frequencies, typically computed for a sample of *n* haploid genomes. Under the assumptions of the Standard Neutral Model (SNM)—including constant population size and panmixia—the expected SFS, averaged across the tree space, is given by $E[\xi_i] = \theta/i$, where $\xi_i$ is the number of sites that carry a derived variant of frequency $i/n$ [8]. The $\theta$ parameter of the SNM is defined as $\theta = 2pN\mu$, where $p$ is the ploidy (typically 1 or 2), $N$ the (effective) population size, and $\mu$ the mutation rate.

Observed SFS in natural populations are often poorly fit by this expectation, owing to violations of one or more of the underlying assumptions of the SNM, including varying population sizes, population structure, direct selection, and linkage with selected sites [9]. A standard procedure in population genetics is thus to first statistically test for the SNM (treated as $H_0$, a null statistical model) and then, when rejected, fit a variety of alternative demographic and/or selection models.

In this article, we show that among a collection of genome-wide SFS from a diverse set of species, many show an unexpected excess of low and high frequency variants, resulting in a U-shaped SFS. Many possible factors may result in such a pattern of variation. These include recent migration from non-sampled populations [10], population structure [11], misorientation of ancestral and derived alleles [12], biased gene conversion [13], recent positive selection at many targets across the genome [14], background selection [15, 16], temporally-fluctuating selection [17], and various reproductive strategies [18].

A number of these scenarios result in an important general violation of Kingman assumptions: the presence of multiple mergers in genealogies (*i.e.*, a node with more than two descendants). Under such scenarios, these distributions are better described by a more general class of models known as the Multiple Merger Coalescent (MMC) [19–23]. Briefly, MMCs may arise when the number of offspring per individual has very high variance, even up to the order of the total population size. Such effects of concentrations of ancestrality (resulting in polytomies in the trees) have been reported in various species across all kingdoms of life [24], and MMC-like genealogies have been observed for species ranging from bacteria (e.g. for *Mycobacterium tuberculosis* [25, 26]) to viruses (e.g. for influenza [27]) to animals (e.g. for the nematode *Pristionchus pacificus* [28], multiple fish species, e.g. [29–31]) and even to cancer cells [32].

Compared to the Kingman coalescent, MMC trees have different distributions of both the branch lengths and the number of lineages that coalesce in each node. They mostly occur when the distribution of offspring numbers is highly variable: a recurring ancestor of a

substantial fraction of the population for the Ψ-coalescent or less specifically an inflated variance for the Beta-coalescent. The most extreme scenario is the presence of a single ancestor for the whole population, resulting in a star-shaped tree where a single node collapses all branches. MMC trees tend, like all star-like trees (e.g. Kingman-like trees in expanding populations), to have an excess of low frequency variants (e.g. derived singletons). Furthermore, the root MRCA node of MMC trees is more often imbalanced than it is for Kingman trees. Imbalanced trees nodes have most leaves on one side while few on the other. As a consequence, MMC trees also display an excess of ancestral rare alleles (e.g. ancestral singletons). Both effects jointly produce a U-shaped SFS (for more details refer to A.1 in S1 Appendix).

Multiple neutral and selective processes can produce MMC genealogies in natural populations. Generally, the term sweepstake reproduction has been proposed for species that have rare individuals with a high reproduction rate coupled with high early-life mortality. In these species, a single or few individuals can become ancestors of a substantial fraction of the population by chance, thus resulting in MMC genealogies (for a review, see [33]). Multiple models featuring the recurrent and rapid emergence of genotypes with high fitness also result in MMC genealogies, often modeled by the Bolthausen-Sznitman coalescent or related models, e.g. [34–38]. Importantly, other biological factors can also lead to MMC-like genealogies, including large rapid demographic deviations [39], seed banks [40], extinction-recolonisation in metapopulations [41] and range expansions [42]. Yet, the frequency of MMC genealogies in nature, and more generally whether MMC models ought to be employed as a more appropriate null for certain species, remains an open question.

In this study, we collected SFS from 45 species (Table 2) from across the tree of life (bacteria, plants, invertebrates and vertebrates), for which genome-wide polymorphism data (with sample sizes of $n \geq 10$) were available together with an outgroup to assign ancestral and derived states. We show that MMC genealogies provide a better fit than the Kingman coalescent in many cases, even when both are combined with non-constant demography and misorientation of ancestral and derived alleles. For several species, the fit is excellent. For each species, we tested two simple MMC models: Beta-MMC [43] and Psi-MMC [44], both tuned by a single parameter that interpolates between a star-shaped tree (*i.e.* a single radiation) to a Kingman-like tree. Demography is here tuned by a single parameter (a simple exponential growth), as is the frequency of misorientation errors. Using composite-likelihood maximization [45] on genome-wide data, we explore statistical power to distinguish between these contributing factors. Finally, we discuss how MMCs may be better utilized in future population genetic analysis, and what evolutionary forces may contribute to the pervasive observation of U-shaped SFS.

## 2 Materials and methods

### 2.1 Coalescent and allele misorientation models

We compared the empirically observed SFS to the theoretical SFS expected under a variety of models. The genealogical models emerge from a discrete generation reproduction model. Each is a (random) tree with $n$ leaves which approximates the genealogy for a sample of size $n$ in a reproduction model in which the population size $N$ is very large ($N \rightarrow \infty$). One unit of time in the coalescent tree corresponds to many generations in the underlying reproduction model: for Kingman's coalescent one time unit corresponds to $N$ generations of a haploid Wright-Fisher model, or order of $N^2$ time steps of an haploid Moran model. This correspondence affects how population size changes are reflected in the coalescent approximation (see definition below, for mathematical justification and details see [46–48]). On the genealogical tree, mutations are placed randomly via a Poisson process with rate $\theta/2$.

We compared three coalescent models: Kingman's $n$-coalescent, Psi-$n$-coalescent (also called Dirac-$n$-coalescent) with parameter $\Psi \in [0, 1]$ and Beta$(2 - \alpha, \alpha)$-$n$-coalescent with $\alpha \in [1, 2]$. The parameters $\alpha$ or $\Psi$ regulate the strength and frequency of multiple mergers: the smaller $\alpha$ or the larger the $\Psi$, the more frequently coalescence events are multiple mergers of increasing size. Both MMCs incorporate Kingman's $n$-coalescent as a special case ($\alpha = 2$ or $\Psi = 0$).

Both MMC coalescent models can be defined for demographic variation that stays of the same order, *i.e.* where the populations size ratio $v_t = N_t/N_0$ of the population size at time $t$ in the past (in coalescent time units) is positive and finite (for large population sizes $N$). The coalescent merges any $k$ of $b$ (ancestral) lineages present at a time $t$ with rate

$$\lambda_{n,k}(t) = v(t)^{-\eta} \int_0^1 x^{k-2}(1 - x)^{n-k} \Lambda(dx), \tag{1}$$

where

- $\Lambda$ could be any probability distribution on [0, 1] but is here either the Dirac distribution (point mass) in $\Psi$ (Psi-coalescent) or the Beta$(2 - \alpha, \alpha)$ distribution (Beta coalescent).

- $\eta$ is a scaling factor reflecting how many time steps from the discrete reproduction model form one unit of coalescent time. More precisely, it is the power of $N$ of the scaling factor: e.g. $\eta = 2$ for a Moran model and $\eta = 1$ for a Wright-Fisher model.

A common way of constructing the $\Lambda$-coalescent, which provides a nice interpretation of Eq (1), is the paintbox process [20]: at rate $x^{-2}\Lambda(dx)$ per time unit, paint each lineage independently with probability $x$ and merge all painted lineages simultaneously. Note that when $\Lambda$ is the Dirac mass at 0, $\lambda_{n,k}(t)$ is nonzero only when $k = 2$, recovering Kingman's coalescent.

We focused on exponentially growing populations, *i.e.* a population size ratio $v(t) = \exp(-gt)$ for growth rate $g \geq 0$ (see A.2 in S1 Appendix for interpretation of $g$ in the initial reproduction model). As underlying reproduction models, we use modified Moran models [44, 47, 49]. At each time step, in a population of size $N$, a single random individual has $U + G$ offspring while $N - U$ random individuals have 1 offspring (leaving $U - 1$ individuals devoid of offspring). As a consequence, the population grows from $N$ to $N + G$ individuals and $G$ is chosen to fit the desired growth rate.

In a standard Moran model, $U = 2$ and $G = 0$, leaving the population size constant. However, for both MMCs, $U$ is set to different values. In both cases, the mean of $U$ does not grow indefinitely with $N$ (for all parameters $\alpha$ and $\Psi$), but the resulting variance does (for $\alpha \neq 2$ and $\Psi \neq 0$).

- In the Psi-$n$-coalescent (essentially [44, 47]), we have $U = 2$, except when a sweepstake event occurs with a small probability of order $N^{-\gamma}$ ($1 < \gamma \leq 2$); in this case, $U = \lfloor N\Psi \rfloor$. In the coalescent time scale, one unit of time corresponds to an order of $N^{\gamma}$ time steps; this is the expected time to a sweepstake event so that $\eta$ must equal $\gamma$. We chose $\gamma = \eta = 1.5$ for $\Psi > 0$, and $\gamma = \eta = 2$ for $\Psi = 0$ (standard Moran model) with $U = 2$ in every time step.

- In the Beta-$n$-coalescent [48, 49], $U$ has distribution $P(U = j) = \lambda_N^{-1} \binom{N}{j} \frac{B(j-\alpha, \alpha+N-j)}{B(2-\alpha, \alpha)}$, where $B$ is the Beta function and $\lambda_N$ is the normalizing constant. Consequently, although the random variable $U$ has a finite mean of at most $\frac{\alpha}{\alpha-1}$, it can take large values with high probability when $\alpha < 2$. See A.2 in S1 Appendix for more details. On the coalescent time scale, one unit of time corresponds to an order of $N^{\alpha}$ time steps, so $\eta = \alpha$. Note that $\alpha = 2$ is the classical Moran model and thus leads to Kingman's coalescent. We stress that allowing for a

exponentially growing population by setting $G > 0$ in the models above does neither change the order of the time scaling nor the resulting coalescent model, it only introduces a slight further rescaling of time in the coalescent, as reflected in the coalescence rates (Eq. 1).

For statistical inference, we treat the observed SFS of $s$ mutations as $s$ independent multinomial draws from the expected SFS (see [45] and [50, Eq. 11] [47, Eq. 14]). This computes an approximate composite likelihood function of the data for any combination of growth rate ($g$) and coalescent parameter ($\alpha$ or $\psi$). However, to include the effect of misorienting the ancestral allele with the derived allele, we introduced another parameter $e$. On average, a misorientation probability of $e$ lets a fraction $e$ of the derived allele carried by $i$ sequences to be falsely seen as appearing in $n - i$ sequences. Additionally, as described in [51, Section 4.2] or [12, p. 1620], as misorientation stems from double-mutated sites, $e$ also relates to the number of sites that cannot be oriented when compared with the outgroup owing to the presence of a third allele (see A.4 in S1 Appendix). We account for these two effects of $e$ by swapping a fraction $e$ of the variants at frequency $i/n$ to $1 - i/n$ and we assume a Jukes-Cantor substitution model [52] to predict for the number $s_{\neq}$ of non-polarizable tri-allelic variants. This leads to a slight variant of [47, Eq. 14]. For any coalescent model with a specific set of coalescent, exponential growth and misorientation parameters, the pseudolikelihood is:

$$PsL(s_1, \ldots, s_{n-1}, s, s_{\neq}) =$$

$$\frac{s!}{s_1! \cdots s_{n-1}!} \prod_{i=1}^{n-1} \left( \frac{E[T_i](1-e) + E[T_{n-i}]e}{E[T_{tot}]} \right)^{s_i} \underbrace{\binom{s + s_{\neq}}{s_{\neq}} \left( \frac{2e}{1+2e} \right)^{s_{\neq}} \left( \frac{1}{1+2e} \right)^s}_{\text{from non-polarizable variants}}, \quad (2)$$

where $s_1, \ldots, s_{n-1}$ is the observed SFS (so we observe $s_i$ sites with derived allele frequency $i/n$), $s = \sum_i s_i$ is the total number of polarizable polymorphic sites and $s_{\neq}$ is the number of non-polarizable sites. $E[T_i]$ is the expected sum of branch lengths that support $i$ leaves in the genealogy and $E[T_{tot}]$ is the sum of all branch lengths. For $e = 0$, we set the term estimated from non-polarizable variants to 1. See A.4 in S1 Appendix for details on the derivation.

## 2.2 Statistical inference

To find the best-fitting parameters, we conduct a grid-search for the highest pseudolikelihood. The expected branch lengths $E[T_i]$ in Eq (2) are computed as in [47], using the approach from [53]. We use the following grids with equidistant steps

**Beta**: $\alpha \in [1, 2]$ in steps of 0.05, $g \in [0, 25]$ in steps of 0.5, $e \in [0, 0.15]$ in steps of 0.01.

**Psi**: $\Psi \in [0, 1]$ in steps of 0.05, $g$, $e$ as for Beta above, complemented with $\Psi \in [0, 0.2]$ in steps of 0.01 (further expanding $g \in [0, 30]$ by steps of 0.5 and $e \in [0, 0.2]$ by steps of 0.01) when $\Psi$ was estimated to be close to 0.

To perform model selection between the three coalescent models, we computed the two following log Bayes factors:

$$BF_1 = \max(\log \max_{\alpha, g, e} PsL, \; \log \max_{\Psi, g, e} PsL) - \log \max_{\alpha=2, g, e} PsL, \quad (3)$$

$$BF_2 = \log \max_{\alpha, g, e} PsL - \log \max_{\Psi, g, e} PsL \quad (4)$$

from the maximum pseudolikelihoods computed for the three models. We inferred a MMC genealogy when $BF_1 > \log(10)$ and further chose a Beta coalescent or a Psi-coalescent when

(additionally) $BF_2 > \log(10)$ or $BF_2 < -\log(10)$ respectively. These arbitrary thresholds have been extensively tested using simulations (see Results), showing that they empirically point to the right model.

We appreciate that the "Bayes Factors" (BF) are computed here as "log-Likelihood Ratios" (log-LR). Interestingly, any likelihood ratio can be interpreted as a posterior probability ratio, provided that the prior on models is uniform (as it is assumed routinely in Bayesian MCMC sampling) as we do here. Thus, in our case, both denominations are equivalent.

For the best fitting parameter combinations either over the full parameter space or restricted to the Kingman coalescent with growth and allele misorientation (*i.e.*, fixing $\alpha = 2$ or $\Psi = 0$), we assessed the goodness-of-fit of the observed data. First, we graphically compare the observed SFS with the expected SFS, approximated as $\left( \frac{E[T_1]}{E[T_{tot}]}, \ldots, \frac{E[T_{n-1}]}{E[T_{tot}]} \right)$. Second, we quantified the (lack of) fit of the data by Cramér's $V$, a goodness-of-fit measure which accounts for different sample sizes and different numbers of polymorphic sites. See A.6 in S1 Appendix for details.

## 2.3 Data

We collected 45 genome-wide SFS that are described in Table 2 and Table G in S1 Appendix. The collected SFS come from public data sets. For 20 data sets, SFS were extracted from whole genome SNP data, including both coding and non-coding regions. For 16 data sets, they were extracted only from transcriptomes (equivalent to coding regions). For 9 bacterial data sets, the SFS were extracted from the core genome. The supplementary material S1 and S2 Figs provide the shapes of the empirically-observed SFS.

# 3 Results

We first demonstrate the power of the methodology using extensive simulations, and then apply it to 45 real SFS computed from a very large variety of taxa.

## 3.1 Statistical performance

Using simulations, we first assess the power of the method to retrieve the correct model and then its power to estimate the parameters. Briefly, for each simulation, we simulated 100 independent loci for each parameter combination, choosing different values for the coalescent parameter ($\alpha$ or $\Psi$), the growth rate of the demographic model ($g$), and the misorientation probability ($e$). For each locus, we then simulate SNPs under an infinite sites model, with a mutation rate such that on average 50 sites are segregating for each locus. This simulation setup is described in further detail in A.7 in S1 Appendix.

Applied to the simulated data, our method performs well. Even for small datasets ($n = 25$), the model selection approach based on Bayes factors computed from Eq (2) identifies the correct multiple merger model in most cases (Table 1), as long as multiple mergers occur with reasonable frequency. As the rate of multiple mergers becomes very low ($\alpha \approx 2$ or $\Psi \approx 0$), mis-identifications are more common (Table 1). However, even when our model prefers the beta-coalescent for data simulated with $\alpha = 2$, in 96% of such cases (with $n = 100$; 71% with $n = 20$), we estimate $\alpha \geq 1.9$, suggesting that even when model mis-idenfication occurs, parameter estimation remains reliable (Table C in S1 Appendix). Over the range of parameter combinations, larger sample sizes lead to smaller errors, as expected. This selection approach is conservative with respect to departures from the standard Kingman coalescent, as we choose a Kingman genealogy model if the Bayes factor does not distinctively point towards an MMC model.

**Table 1. Model selection via two-step Bayes factor criterion.** Based on 2,000 simulations for each true model assuming $n = 25$ individuals with 100 loci with 50 mutations each on average. For each simulation, the coalescent parameter is fixed and the growth parameter $g$ and the allele misorientation rate $e$ are randomly chosen ($g \in [0, 11.25]$, $e \in [0, 0.1]$). The second column shows whether the parameters used for simulation were included in the inference grid. Fractions are rounded to two digits. The maximum of each row is marked in bold. MMC refers to cases in which neither the Psi- nor Beta-coalescent is preferred. An expanded version with enhanced sample size is provided in Table B in S1 Appendix. For details on simulations and inference parameters see A.7 in S1 Appendix.

| True model | Within the grid? | Fraction model inferred as | | | |
|---|---|---|---|---|---|
| | | Kingman | Beta | Psi | MMC |
| $\alpha = 2$ | yes | **0.79** | 0.21 | | |
| $\alpha = 1.9$ | yes | 0.34 | **0.66** | | |
| $\alpha = 1.8$ | yes | 0.02 | **0.91** | 0.04 | 0.03 |
| $\alpha = 1.625$ | no | | **0.9** | 0.06 | 0.05 |
| $\alpha = 1.025$ | no | | **1** | | |
| $\Psi = 0.005$ | no | **0.55** | 0.45 | | |
| $\Psi = 0.025$ | no | 0.05 | **0.72** | 0.14 | 0.09 |
| $\Psi = 0.05$ | yes | | 0.12 | **0.82** | 0.06 |
| $\Psi = 0.075$ | no | | 0.06 | **0.91** | 0.03 |
| $\Psi = 0.1$ | yes | | 0.02 | **0.98** | |

Parameter estimation within both the Beta- and Psi-coalescent models works well for multi-locus data for large enough samples, especially for the allele misorientation rate $e$ and for the coalescent parameter $\alpha$ or $\psi$ (Fig 1 and Figs A.4–A.6 in S1 Appendix). The growth rate, in contrast, is only estimated well for situations where the simulated growth rate was low (Figs A.11, A.14, A.17 and A.20 in S1 Appendix).

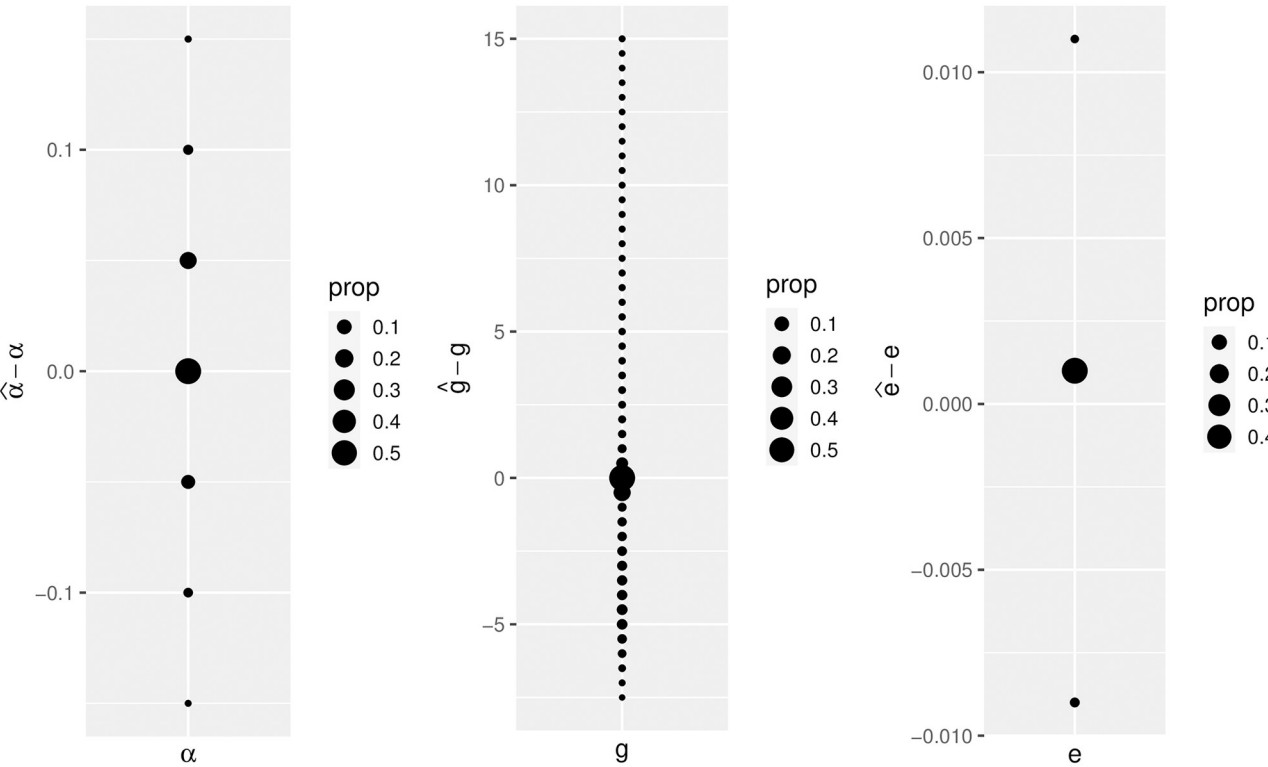

**Fig 1. Error for estimating parameters for Beta coalescents with exponential growth and allele misorientation across the parameter grid for ($\alpha$, $g$, $e$).** The space between the points stems from the grid. Sample size $n = 100$, 50 independent loci with 100 mutations on average. 500 simulations were performed per parameter triplet.

### 3.2 Data analysis

The simulations demonstrate that the method is able to retrieve the correct model, and also correctly estimate the parameters of the MMC, provided that there is enough signal in the data. Next, we applied the method to 45 real SFS from 45 distantly related taxa. We first tested how many datasets are better fit by an MMC model than by a Kingman model, then tested the goodness of the MMC fit and estimated MMC parameters for real data.

**MMC fits better than Kingman.**   First, we assessed the fit of each SFS to both MMC models and the Kingman coalescent, with exponential growth and misorientation. Using the Bayes Factor criterion, we selected the best fitting model for each empirical SFS in our dataset (Table 2). A large majority (76%) of the SFS produce a better fit to MMC models than to the standard Kingman coalescent model. The best model is most frequently the Beta-coalescent (53%), followed by the Kingman coalescent (24%) and the Psi-coalescent (13%). In a few cases, both MMC models produce a better fit than the coalescent, but we cannot distinguish the best fitting MMC (9%).

**MMC is sometimes a good fit.**   While we show that MMC models produce better fits than the Kingman coalescent across many species, this could be because no model fits well. To test whether the best fit coalescent model is indeed a good model to predict the observed SFS, we calculated Cramér's $V$, a measure of goodness-of-fit appropriate for variable contingency tables (e.g., SFS with different sample sizes across species, see A.6 in S1 Appendix for details). Combined with visual inspections (all SFS with their fit are provided in supplementary material S1 and S2 Figs), we designed empirical grade categories from 'very accurate' fit to 'very poor' fit, as following: A: $V \in [0 : 0.033[$, B: $V \in [0.033 : 0.066[$, C: $V \in [0.066 : 0.1[$ and D: $V \in [0.1 : \infty[$. Importantly, the MMC models fit well to 67% of data sets: 30/45 SFS have grades A or B on Table 3. This demonstrates that not only is MMC a better choice than Kingman on statistical grounds but also that it appears as a good model to predict patterns of diversity for a large majority of species.

**The amount of multiple mergers greatly varies among species.**   The MMC models we use vary in the extent of multiple mergers, from star-like to Kingman-like, scaled by a single parameter ($\alpha$ and $\Psi$ respectively for the Beta- and Psi-coalescent). To determine whether the model fits suggest an appreciable level of multiple mergers, we next explore the estimated parameters for MMC models. Of the 45 empirical SFS we analyzed, 73% (33/45) have $\hat{\alpha} < 1.9$ under the Beta-coalescent, which suggests a non-trivial frequency of multiple mergers, and implies something that is not captured by the SNM is occurring in these species (see Table D in S1 Appendix for $\alpha$ estimates of all data sets, including those where the Kingman or Psi-coalescent are the best fit model). Nonetheless, estimates of $\alpha$ and $\Psi$ are both skewed towards values that approach the Kingman coalescent (2 and 0, respectively), despite covering the full range of values across the tree of life (Fig 2 and Fig A.9 in S1 Appendix).

**Assuming a Kingman coalescent leads to an overestimation of the growth rate.**   One potential impact of using the standard Kingman coalescent instead of better-fitting MMC models is the incorrect estimation of other parameters, including aspects of demography. To explore this issue, we compared the estimated growth rate and misorientation error assuming a Kingman model rather than an MMC model. We observe that the growth parameters are often higher when inferred under the the Kingman coalescent than in either of the MMC models (Table D in S1 Appendix), although estimates of $g$ tend to converge in empirical datasets where MMC parameter estimate approaches Kingman (Fig A.7A in S1 Appendix). This mirrors previous results of compensating the effect of MMC when inferring under a Kingman coalescent by estimating a higher growth rate in our scenario without allele misorientation, see e.g. [47].

**Table 2. Data sets description: Taxa, Species, number of haplotypes (*n*) and number of polymorphic sites (#SNP).** Best fitting model (Kingman (KM), Beta, Psi-coalescent or no preference between Multiple Merger Coalescents (MMC)), its parameters (parameters describing coalescence (Coal), growth rate (*g*) and misorientation (*e*)) and goodness-of-fit grade from Cramér's *V* values.

| Order | Species | *n* | #SNP | Model | Coal | $g_{Model}$ | $e_{Model}$ | Grade |
|-------|---------|-----|------|-------|------|-------------|-------------|-------|
| Vertebrates | *Aptenodytes patagonicus* | 20 | 1,278 | Beta | 1.25 | 1.5 | 0 | B |
| | *Athene cunicularia* | 40 | 11,268,203 | Beta | 1.8 | 1 | 0.03 | B |
| | *Corvus cornix* | 38 | 7,551,159 | Beta | 1.85 | 0.5 | 0 | A |
| | *Coturnix japonica* | 20 | 5,061,864 | Beta | 1.45 | 0.5 | 0.01 | A |
| | *Egretta garzetta* | 10 | 9,318,499 | Beta | 1.75 | 0 | 0.02 | B |
| | *Emys orbicularis* | 20 | 515 | KM | ∅ | 0.5 | 0 | C |
| | *Ficedula albicollis* | 24 | 14,697,230 | Ψ | 0.01 | 0.5 | 0.01 | A |
| | *Gorilla gorilla gorilla* | 54 | 9,878,547 | Beta | 1.9 | 0 | 0 | B |
| | *Homo sapiens* | 216 | 19,441,528 | Beta | 1.85 | 0 | 0 | A |
| | *Lepus granatensis* | 20 | 769 | MMC | 0.12 | 0 | 0.03 | C |
| | *Nipponia nippon* | 16 | 1,140,694 | KM | ∅ | 0 | 0.03 | D |
| | *Pan paniscus* | 26 | 6,293,657 | Beta | 1.85 | 1 | 0 | B |
| | *Pan troglodytes ellioti* | 20 | 10,009,190 | Beta | 1.7 | 0 | 0 | A |
| | *Parus major* | 54 | 14,174,305 | Beta | 1.75 | 0 | 0.01 | A |
| | *Parus caeruleus* | 20 | 866 | MMC | 0.04 | 0 | 0.02 | B |
| | *Passer domesticus* | 16 | 18,501,992 | KM | ∅ | 0 | 0 | A |
| | *Phylloscopus trochilus* | 24 | 33,401,127 | KM | ∅ | 12.5 | 0 | A |
| | *Taeniopygia guttata* | 38 | 53,263,038 | Beta | 1.75 | 4 | 0 | A |
| Invertebrates | *Armadillidium vulgare* | 20 | 23,323 | Beta | 1.7 | 0 | 0.03 | C |
| | *Artemia franciscana* | 20 | 5,548 | Beta | 1.65 | 0 | 0.03 | B |
| | *Caenorhabditis brenneri* | 20 | 1,339 | Beta | 1.5 | 0 | 0.06 | C |
| | *Caenorhabditis elegans* | 574 | 165 | KM | ∅ | 0 | 0.06 | D |
| | *Ciona intestinalis A* | 20 | 1491 | Beta | 1.9 | 0 | 0.03 | C |
| | *Ciona intestinalis B* | 20 | 2186 | Beta | 1.6 | 0 | 0.02 | C |
| | *Culex pipiens* | 20 | 5,442 | Beta | 1.55 | 0.5 | 0.01 | B |
| | *Drosophila melanogaster* | 196 | 4,662,706 | Beta | 1.65 | 0.5 | 0.02 | A |
| | *Halictus scabiosae* | 22 | 712 | MMC | 0.04 | 0 | 0.01 | B |
| | *Melitaea cinxia* | 18 | 1,695 | Beta | 1.7 | 0.5 | 0.03 | B |
| | *Messor barbarus* | 20 | 9,651 | KM | ∅ | 0.5 | 0 | C |
| | *Ostrea edulis* | 20 | 939 | MMC | 0.04 | 0 | 0.02 | B |
| | *Physa acuta* | 18 | 4,286 | Beta | 1.5 | 0 | 0.02 | B |
| | *Sepia officinalis* | 18 | 1,740 | KM | ∅ | 0 | 0.02 | C |
| Plants | *Arabidopsis thaliana* | 345 | 10,322,757 | Beta | 1.6 | 0 | 0.07 | A |
| | *Zea mays* | 66 | 520,310 | Ψ | 0.01 | 0 | 0 | A |
| Bacteria | *Acinetobacter baumannii* | 79 | 78,175 | Beta | 1.8 | 0 | 0.1 | B |
| | *Bacillus subtilis* | 38 | 105,523 | Ψ | 0.14 | 0 | 0.2 | B |
| | *Chlamydia trachomatis* | 59 | 9,924 | KM | ∅ | 0 | 0.11 | D |
| | *Clostridium difficile* | 11 | 192 | KM | ∅ | 15 | 0.15 | D |
| | *Escherichia coli* | 62 | 84,222 | KM | ∅ | 0 | 0.06 | B |
| | *Helicobacter pylori* | 70 | 27,498 | Ψ | 0.01 | 1 | 0.2 | B |
| | *Klebsiella pneumoniae* | 156 | 203,601 | KM | ∅ | 18.5 | 0.15 | D |
| | *Mycobacterium tubercolosis* | 33 | 7,142 | Beta | 1.05 | 2.5 | 0 | C |
| | *Pseudomonas aeruginosa* | 86 | 90,258 | Ψ | 0.06 | 3 | 0.2 | B |
| | *Staphylococcus aureus* | 152 | 30,052 | Ψ | 0.01 | 1 | 0.2 | B |
| | *Streptococcus pneumoniae* | 32 | 49,917 | Beta | 1.5 | 0 | 0.08 | C |

**Table 3. Distribution of goodness-of-fit grades of the best-fitting models for the 45 collected SFS.** Calculated from Cramér's $V$, A: $V \in [0 : 0.033[$, B: $V \in [0.033 : 0.066[$, C: $V \in [0.066 : 0.1[$ and D: $V \in [0.1 : \infty[$.

| Model \ Grade | A | B | C | D | Total |
|---|---|---|---|---|---|
| Kingman | 2 | 1 | 3 | 5 | 11 |
| Beta | 8 | 10 | 6 | | 24 |
| Psi | 2 | 4 | | | 6 |
| MMC | | 3 | 1 | | 4 |
| Total | 12 | 18 | 10 | 5 | 45 |

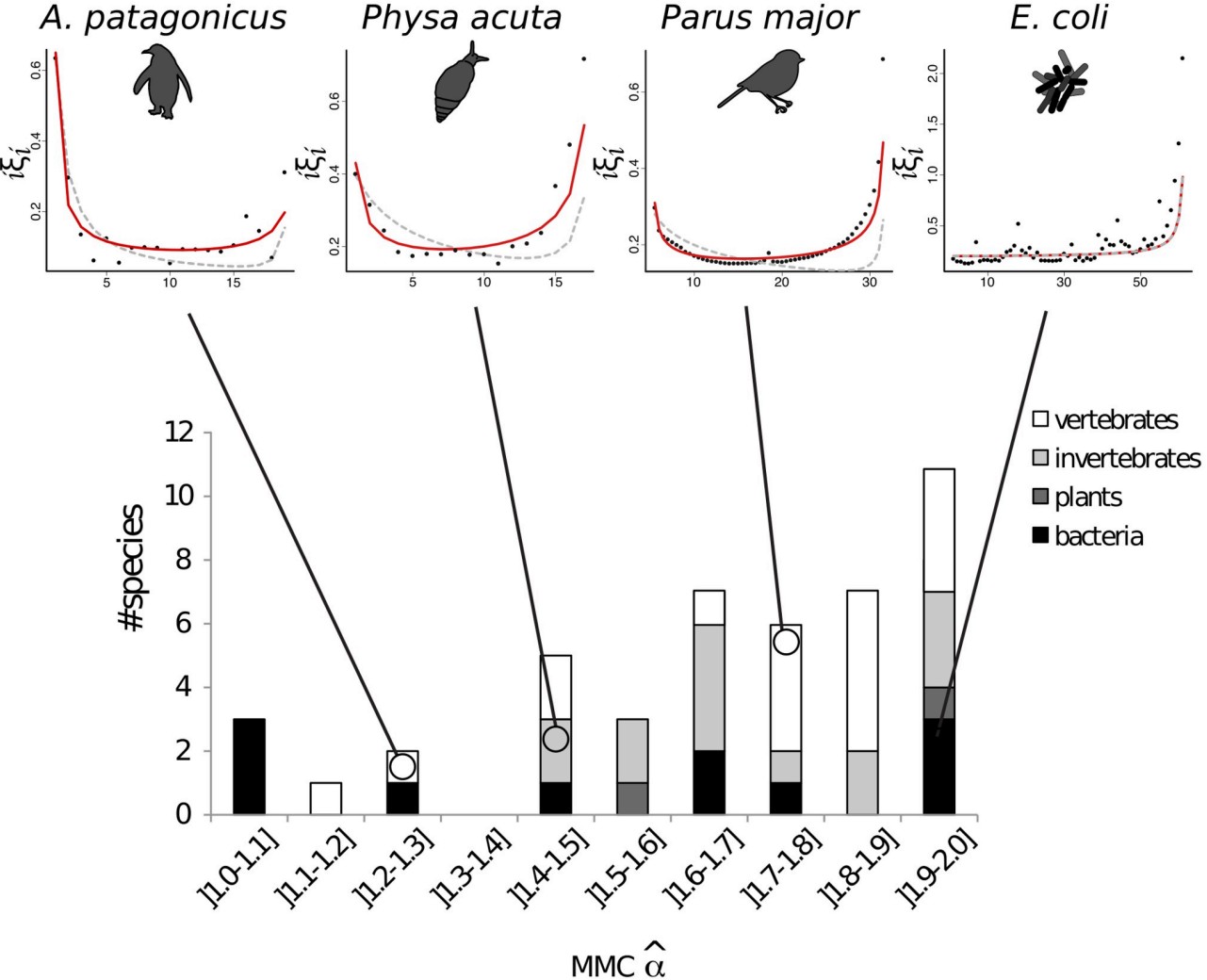

**Fig 2. Estimates of $\alpha$ by species.** The four top panels represent transformed $\phi$-SFS ($\phi_i = i\xi_i$ as in [54, 55]) for four species from different taxa: two vertebrates *Aptenodytes patagonicus* (left) and *Parus major* (center right) an invertebrate *Physa acuta* (center left), and a bacteria *Escherichia coli* (right). For *E. coli*, the uptick in the spectrum comes exclusively from the allele miss-orientation, as $\hat{\alpha} = 2$. Black dots are the observed values, grey dotted lines are the best fits under the Kingman's coalescent model and red lines are the best fits under a Beta-coalescent model.

In contrast, the allele misorientation parameters $e$ are almost identical between the Kingman model and the MMC (Fig A.7B in S1 Appendix), which may be a consequence of adding a second, coalescent-model-free estimation method for $e$ to the pseudolikelihood 2. This suggests that for datasets with frequent multiple mergers, assuming a Kingman model may lead to overestimating $g$, but is not likely to impact estimates of $e$.

**Both MMC models have similar parameter estimates.**    Finally, we compare the estimations of both MMC models to see whether using one or the other would result in qualitatively different conclusions. The parameters inferred under the two MMC models are highly correlated. The multiple merger parameters $\alpha$ of the Beta-coalescent and $\Psi$ of the Psi-coalescent are negatively correlated, as expected from their definitions (Fig A.8A in S1 Appendix, Spearman correlation: $\rho = -0.72$). The estimated growth and misorientation parameters are highly positively correlated (Spearman correlations $\rho = 0.73$ and $\rho = 0.95$). The case of *Clostridium difficile* is a notable exception. The best model inferred is the Kingman, consistent with $\hat{\Psi} = 0$ inferred for the Psi-coalescent, but for the Beta-coalescent $\hat{\alpha} = 1$, the strongest MMC component, is estimated. However, this discrepancy is likely due to statistical noise: the data set is very small (192 mutations in a sample size $n = 11$) and the species has a very low recombination rate.

## 4 Discussion

In this study, we show that unfolded SFS for large variety of species show a characteristic U-shape, which is inconsistent with the expectations of the standard neutral model using the Kingman coalescent. One possible explanation for this observation is the prevalence of MMC and MMC-like genealogies in real populations. To explore the role of MMCs in these data, we develop a statistical framework to detect MMC models. Using simulated data, we show this approach has power to detect the correct MMC model and estimate its parameters, provided that the data are informative enough. Using real SFS collected from 45 species across the tree of life, we further show the MMC models are a better fit than the Kingman coalescent in most species, even when population growth and orientation errors are additionally modeled, although in some cases the MMC parameter suggests approximately Kingman behavior. In the following, we discuss some possible biological implications of these observations.

### Chosen multiple-merger models, alternatives and limitations

We chose two commonly used haploid multiple-merger models, the Beta- and the Psi-coalescent, which were previously associated with sweepstake reproduction in the literature [43, 44]. However, these MMC models may also originate either from alternative neutral processes or from selective processes. Indeed, the Beta $n$-coalescent with $\alpha = 1$ is known as the Bolthausen-Sznitman $n$-coalescent and it (resp. a slight variant of it) emerges in a variety of models with rapid selection [34–38]. The Beta-coalescent has also been associated with range expansions [42]. In addition, Psi $n$-coalescents have been successfully used as proxy models for detecting regions experiencing positive selection [56].

While Beta- and Psi-coalescent models are linked to several biological properties potentially present in a considerable number of species, these are not the only MMCs used to model biological populations. For instance, in the modified Moran models presented above, one can let the $\Psi$ be random, leading to another more general class of MMC that also belongs to the family of $\Lambda$-coalescents [49], which is a generally good candidate for sweepstakes reproduction. Other alternative models exist that more closely mimic recurrent selective sweeps [57] or appear as variants of Psi- and Beta- coalescents, but for diploid reproduction [58–60].

We have chosen to evaluate two simple classes of coalescent processes which interpolate between the two extreme tree shapes—a purely bifurcating Kingman tree ($\Psi = 0$ or $\alpha = 2$) and

a star-shaped tree ($\Psi = 1$ or $\alpha = 0$). Alternative multiple merger models could potentially be (mis)identified as Beta- or Psi-coalescents, as previously shown [61]. Our method should thus still be able to detect multiple merger signals even if caused by processes that lead to another MMC. Assessing further which MMC models are best fitting for biological populations could be informative [26]. In this regard, our inference approach is based on computing $E(T_i)$ from Eq (2) via the method from [53], so it can easily be extended to incorporate most multiple merger models (any $\Lambda$- or $\Xi$-coalescent) and any demographic histories, by replacing the Markov transition rate matrix of the coalescent and the population size profile $v$.

To assess the quality of our inference method, we used a simplified approach where unlinked loci are assumed to be independent. This is not always true for MMC models (see [62] and A.10 in S1 Appendix), especially for Psi-coalescents caused by strong sweepstake reproduction events with $\Psi$ well above 0. Thus, the real error rates of our techniques could be higher than anticipated by our simulation study. However, this potential increase in error rates can be offset by the presence of datasets that are larger than those assumed in our simulation study. Additionally, due to our reliance on the expected SFS entries—which are averages over the tree space—our inference method (and also our goodness-of-fit assessment) should perform worse (given identical sample sizes and mutation counts) when used on species with small genomes and low recombination rates. This tendency is clearly visible in the goodness-of-fit tests of multiple bacterial data sets.

## Non-extreme demography alone cannot generate U-shaped SFS

The Kingman coalescent for a population undergoing non-extreme demographic changes corresponds simply to a monotonic time rescaling of the standard Kingman coalescent. Non-extreme changes mean that the population size changes occur at the same time scale than coalescent time. For the MMC models employed, this is for instance satisfied if the population size stays of the same order (N) throughout generations. If this is true, changes in population size correspond to changes in waiting times, but not topology, of the tree. The expected SFS for a large population and a large sample is a linear function of the expected waiting times $c_k$ for the next coalescence of $k$ lineages, with a simple analytical form:

$$E[\xi_f] = \theta \sum_{k=2}^{\infty} k(k-1)c_k \cdot (1-f)^{k-2},$$

(5)

where $\xi_f$ is the number of variants at frequency $f$. Since the expected waiting times are positive $c_k > 0$, all coefficients in this expansion are positive. This means that the spectrum has a positive value, negative derivative, positive convexity (second derivative), etc., so it is a completely monotonic function ('no bumps'). A similar argument holds for finite frequencies $i/n$ [63]. More details are provided in A.13 in S1 Appendix. As it is monotonically decreasing with $i$, U-shaped spectra cannot occur as a result of any non-extreme demographic dynamics alone. Note however that extreme changes in population size violate this and may lead to multiple merger genealogies [39, 64].

## Alternative processes leading to *U*-shaped SFS, further confounding factors

Our model directly incorporates MMC genealogies, exponential growth combined with allele misorientation as sources of the U-shape of the SFS. However, other potential factors can also influence the SFS and produce SFS with similar shapes. We further discuss here three particularly notable factors, further sources of misorientation errors, population structure (e.g. gene flow or admixture) and biased gene conversion.

First, we tested whether other sources of misorientation errors can explain the strong support for MMC in the dataset. As sequencing errors in the in-group will likely create mostly derived singletons, they cannot explain the U-shape. Furthermore sequencing errors in the outgroup would result in the exact same patterns as natural mutations. Thus the amount of misorientation error can include both recurrent mutations and sequencing errors in the outgroup. We have also developed an extended version of the orientation errors model (see A.4.1 in S1 Appendix) taking into account different rates for transitions and transversions [65]. Even though orientation errors are then modeled by two parameters, the general picture is the same: the best supported model (Tables E and F in S1 Appendix) remains unchanged for 33 species and becomes another MMC for 6 species. However, 6 species have their statistical support swapped between an MMC and Kingman models: *A. franciscana*, *C. cornix*, *F. albicollis*, *M. cinxia*, *P. paniscus* (Beta to Kingman for the 5) and *K. pneumoniae* (Kingman to Beta), leaving 66% of the species with a better support for MMC than Kingman. We then tested whether the phylogenetic proximity of the outgroup could allow for Incomplete Lineage Sorting (ILS) that can cause ancestral polymorphisms to segregate in the sampled species (see A.5 in S1 Appendix). Results (Table H in S1 Appendix) show that ancestral polymorphisms (ILS with mutation) is not a likely contributor of orientation errors for most species as they cannot represent an appreciable amount of the polymorphic sites. However, for the 10 species for which the estimated $P(ILS) \times 0.1$ is larger than 1%, the error rate is possibly underestimated. However, 3/10 show strong support for the Kingman coalescent. Therefore, which model would have the best statistical support for these species if ILS was properly account for in a dedicated non-trivial MMC likelihood framework remains unclear.

Second, to explore population structure, we performed a PCA analysis of all datasets, followed by a *k*-means clustering (results in Table H in S1 Appendix). We acknowledge the possibility that unsampled "ghost" demes can exist and could potentially result in U-shaped SFS [10], and that some cases of metapopulation dynamics results in MMC trees [41, 42]. Assessing the presence of ghost demes from genetic data is challenging. Importantly, among the 11 species that display a clear pattern of genetic structure, only 6 have an observed U-shaped SFS that is well fitted (grades A and B) by an MMC model. Furthermore, among the 14 species with no clear structure, 10 have an observed U-shaped SFS well fitted by an MMC model. This suggests that population structure is not the main cause of the U-shape of the observed SFS. Additionally, many species with clear structure have low goodness-of-fit grades (C and D), suggesting that none of the models we compare are a good fit to these datasets. We however note that 8/11 species with a clear structure pattern are Bacteria. Indeed for the small genomes with low recombination rate (in Bacteria recombination preserves long distance linkage), the apparent structure does not necessarily equate with population structure, but may instead arise from the limited number of genealogies. At the limit, a single Kingman tree would result in a clear structure pattern due its long internal branches.

To check for the effect of biased gene conversion, we built alternative SFS only based on a subset of unbiased mutations that are immune to biased gene conversion (details in A.9 in S1 Appendix, the unbiased SFS are added in supplementary material S1 and S2 Figs). Many of these unbiased SFS were only slightly changed, and many kept their *U*-shape. However 6 species (*A. cunicularia*, *F. albicollis*, *E. garzetta*, *P. maior*, *O. edulis*, *P. troglodytes e.*, all but one vertebrates) lost their U-shape. Two have a small sample size (*E. garzetta*) or a low multiple merger component estimate (*F. albicollis*). For these species, it is nonetheless possible that the U-shape is caused by biased gene conversion.

In a very conservative approach, among the 17 data sets showing robust and strong MMC signals (category A, B in Table 2, with $\alpha \leq 1.8$ or $\Psi \geq 0.04$ and sample size $\geq 20$), 6 cases may arise due to structured genetic diversity (*A. baumannii*, *D. melanogaster*, *H. pilori*, *O. edulis*,

*P. aeruginosa* and *S. aureus*) and 3 more lose their characteristic U-shape when biased gene conversion is accounted for (*A. cunicularia*, *P. maior*, *P. troglodytes*; *O. edulis* being in common). Thus, 8 species have strong support for MMC models with population growth. We believe that at least for these cases (and likely for more), neutral sweepstake reproduction, frequent selection, or other factors that can produce MMC-like genealogies ought to be seriously considered as underlying drivers of their genetic diversity.

Importantly and more generally, among the 30 species that display a good statistical fit (with grades A and B), 27 point to MMC models whereas only 3 point to a Kingman coalescent. Noting that MMC models encompass the Kingman coalescent as a special case, our results support the view that MMC models may often constitute better reference models.

## MMC and biological properties

Although we only analyzed a small number of species sampled non-uniformly across the tree of life, we often observed signatures of multiple merger-like events. Reassuringly, our analysis supports multiple merger genealogies for *Mycobacterium tuberculosis*, which was recently proposed in [25] and [26] (the non-optimal goodness-of-fit likely stems from a small and essentially non-recombining genome). The strongest multiple merger effects estimated within the class of Beta coalescents ($\alpha \leq 1.1$) were found in two bacterial pathogens with low or intermediate recombination rates (*M. tuberculosis* and *P. aeruginosa*). There also does not seem to be a meaningful correlation between MMC effects and overall genetic diversity (Fig A.23 and Table I in S1 Appendix). We stress that links between MMC model parameters and biological properties are not always obvious. For example, while reproduction sweepstakes can lead to both Beta- and Psi-coalescents, it is not straightforward to translate the parameters $\alpha$ and $\Psi$ into realistic offspring distributions. For instance the Psi-coalescent model hypothesizes that an occasional individual contributes a fraction $\Psi$ of the next generation, though examples of such a single-individual contribution are not biologically likely. Still, the coalescent approximations do fit well to data. Importantly, different reproduction models can result in the same model on the *coalescent time scale*. The large families of the MMC models could result from the rapid accumulation of coalescences over multiple generations instead of in a single one.

## Conclusion

We analyzed genomic data for 45 species across the tree of life, and showed that many exhibit a U-shaped SFS. By developing a statistical approach to distinguish the genetic signatures of different potential sources of this U-shape: allele misorientation and MMC genealogies, together with exponential population growth, our results show that while some U-shaped SFS are well-described by only allele misorientation, the majority are better described by models that include an MMC component (27 point to MMC and only 3 to Kingman coalescent, with the rest inconclusive). However, distinguishing true MMC from MMC-like processes remains challenging. For example, both biased gene conversion (evident for 6 species) and population structure (clear for 11 species, many of which had no U-shapes) could also generate U-shaped SFS, and appear to be plausible explanations for the observed data of certain species. MMC models with simple growth nonetheless represent an excellent fit for at least 8 species. More complex demographic scenario (with more parameters) can be included in the MMC framework presented here and can be statistically tested when demographic inference is being performed. However non-extreme variations of population sizes cannot explain the pervasive observation of U-shaped SFS.

This study thus invites both closer inspection for the species at hand, but also suggests that MMC genealogies may appear in a wider range of species than previously reported (e.g., a few

marine species and multiple human pathogens). For such species, their biological properties likely render MMC rather than Kingman models as the more fruitful analysis framework, highlighting the importance of further developing both theory and statistical inference procedures under these lesser-used models [66].

## Supporting information

**S1 Appendix. Extended methods, supplementary figures and tables.**
(PDF)

**S1 Fig. Observed SFS and expected SFS under best fitting models.**
(PDF)

**S2 Fig. Observed transformed SFS and expected transformed SFS under best fitting models.**
(PDF)

**S3 Fig. PCA plots w. DAPC colours.**
(PDF)

**S4 Fig. BIC plots of DAPC population structure analyses.**
(PDF)

**S5 Fig. Per-chromosome PCA+DAPC and BIC plots for D. melanogaster data.**
(PDF)

## Acknowledgments

We would like to thank Allison J Shultz and Brian J Arnold for help with producing VCFs for several bird species. The authors acknowledge the support by the state of Baden-Württemberg (Germany) through bwHPC.

## Author Contributions

**Conceptualization:** Fabian Freund, Elise Kerdoncuff, Sebastian Matuszewski, Marguerite Lapierre, Jeffrey D. Jensen, Luca Ferretti, Amaury Lambert, Timothy B. Sackton, Guillaume Achaz.

**Data curation:** Fabian Freund, Elise Kerdoncuff, Timothy B. Sackton.

**Formal analysis:** Fabian Freund, Amaury Lambert, Guillaume Achaz.

**Investigation:** Jeffrey D. Jensen, Luca Ferretti, Guillaume Achaz.

**Methodology:** Fabian Freund, Elise Kerdoncuff, Sebastian Matuszewski, Marguerite Lapierre, Marcel Hildebrandt, Luca Ferretti, Amaury Lambert, Guillaume Achaz.

**Project administration:** Guillaume Achaz.

**Resources:** Timothy B. Sackton.

**Software:** Fabian Freund, Elise Kerdoncuff, Sebastian Matuszewski, Marcel Hildebrandt, Guillaume Achaz.

**Supervision:** Fabian Freund, Jeffrey D. Jensen, Amaury Lambert, Guillaume Achaz.

**Validation:** Sebastian Matuszewski, Jeffrey D. Jensen, Timothy B. Sackton, Guillaume Achaz.

**Visualization:** Elise Kerdoncuff, Marguerite Lapierre, Guillaume Achaz.

**Writing – original draft:** Fabian Freund, Elise Kerdoncuff, Sebastian Matuszewski, Jeffrey D. Jensen, Luca Ferretti, Amaury Lambert, Timothy B. Sackton, Guillaume Achaz.

**Writing – review & editing:** Fabian Freund, Jeffrey D. Jensen, Timothy B. Sackton, Guillaume Achaz.

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
