## [Decision Letter · Decision Letter 0]

20 Jun 2022

Dear Dr Achaz,

Thank you very much for submitting your Research Article entitled 'Interpreting the pervasive observation of U-shaped Site Frequency Spectra' to PLOS Genetics.

The manuscript was fully evaluated at the editorial level and by independent peer reviewers. All three expert reviewers were enthusiastic about your work, but raised some substantial concerns. Based on the reviews, we will not be able to accept this version of the manuscript, but we would look forward to reviewing a revised version. We cannot, of course, promise publication at that time.

Should you decide to revise the manuscript for further consideration here, your revisions should address each of the specific points made by each reviewer. We will also require a detailed list of your responses to the review comments and a description of the changes you have made in the manuscript.

Three issues that have been raised by the reviewers are particularly important and should be addressed with care in the revision.  The effects of relaxing the Jukes-Cantor assumption (specifically, transition:transversion bias) should be checked in simulation, as suggested by Reviewer 1, or included in the model (Eq'n 2).  The effects of ancestral mis-specification should also be examined (sequencing errors in the outgroup), as should the effects of ancestral variation (SNPs already segregating in the ancestral population).  The latter two issues are identified and explained in greater detail by Reviewer 2.

The reviewers also suggest that the manuscript could benefit from a more intuitive explanation, perhaps with some illustrative diagrams, of the differences underlying the Kingman, beta and psi coalescents, and why the latter two, or MMC in general, produce U-shaped site frequency spectra.  This would also be important in aiming the manuscript toward the relatively broad audience of PLoS Genetics.

If you decide to revise the manuscript for further consideration at PLOS Genetics, please aim to resubmit within the next 60 days, unless it will take extra time to address the concerns of the reviewers, in which case we would appreciate an expected resubmission date by email to plosgenetics@plos.org.

[LINK]

We are sorry that we cannot be more positive about your manuscript at this stage. Please do not hesitate to contact us if you have any concerns or questions.

Yours sincerely,

Lindi Wahl

Associate Editor

PLOS Genetics

Bret Payseur

Section Editor: Evolution

PLOS Genetics

Reviewer's Responses to Questions

**Comments to the Authors:**

Reviewer #1: This study investigated how the empirical allele frequency spectra from many species match the theoretical predictions of multiple merger coalescent (MMC) vs. Kingman’s coalescent. The result that MMC better explains the majority of data has a potentially great impact in the investigation of major evolutionary processes governing the pattern of genetic variation in natural populations. I think this study was carefully conducted and clearly presented. I have only one major comment (#1 below) and other minor comments/suggestions on this manuscript.

1. Using eq. (2) that combines the probability of observing a third allele in the outgroup as a function of the probabilitya of misorientation is a critical step in this study, since a U-shaped spectrum depends sensitively on the frequency of misorientation. However, I wonder the use of this pseudolikelihood function leads to under-estimation of misorientation probability, as the Jukes-Cantor model of DNA sequence evolution was assu med. In many sites where ingroup polymorphisms are due to mutations that are transitions, third alleles in the outgroup arise due to mutations that are transversions. As it is known that transitions occur about five times faster than transversions, then, the probability of double mutations leaving two alleles in the sample can be larger than that leaving three alleles in the sample, while they are assumed p and 2p in the derivation of eq. (2), respectively. Therefore, the inference of less than actual numbers of misorientation, thus overestimation of high frequency derived alleles, probably have been made. I think the effect of violating the Jukes-Cantor assumption can be easily checked in simulation, or eq. (2) can be easily modified to take transition/transversion bias into account.

2. It will be nice if the authors can provide more illustrations from which the readers with little theoretical background can gain intuitive, biological understanding of differences between beta-, psi-, and Kingman’s coalescent. In particular, how beta and psi coalescent disagree with each other is hard to imagine. The best-fitting psi-coalescent spectra might be added to Figure 2. Or, for a given series of Tajima’s D or Fay and Wu’s H values (statistics that geneticists are more familiar with), best matching beta, psi, and Kingman coalescent spectra could be plotted.

3. p.12, paragraph starting with line 13. Not all structured populations may not be identified from the PCA analysis. How about a species found in one area but undergoing metapopulation dynamics, i.e. with recurrent extinction-recolonization of multiple patches within the area?

4. From which population(s) was the D. melanogaster sampled? Since fruit fly data is the most abundant one in animals (along with humans), it might be considered to divide the sample into one from a population of relatively stable demographic history and the other from populations of more deviation from simple demography. Likewise, human samples from populations other than Yoruba (with evidence of rapid expansion) might be tested separately. Such tests might help us understand the effect of demography in the mode of coalescent.

Reviewer #2: Review of "Interpreting the pervasive observation of U-shaped Site Frequency Spectra" by Fabian Freund, Elise Kerdoncuff, Sebastian Matuszewski, Marguerite Lapierre, Marcel Hildebrandt, Jeffrey D. Jensen, Luca Ferretti, Amaury Lambert, Timothy B. Sackton and Guillaume Achaz:

In this manuscript, the authors investigate whether the SFS, a widely-used summary statistics of genome-wide data, of several different species across the tree of life is better fit by a genealogical model of multiple mergers, rather than the widely used standard model Kingman's coalescent, which only allows binary mergers in the genealogy. The authors use the term U-shaped Site Frequency Spectra, since an excess of high frequency derived variants in the SFS is a tell-tale of multiple merger genealogies, as this cannot be observed under Kingman's coalescent.

To this end, the authors develop an inference framework, to infer the best fitting parameters under Kingman's coalescent, the Psi-coalescent, and the Beta-coalescent, the latter two are two widely used 1-parameter families of multiple merger genealogies, which include Kingman's coalescent at one end of the parameter range. The authors include two nuisance parameters in the framework to account for confounding factors. First, they include a parameter for a model of exponential growth (or decline), since the demographic history of the population does affect the SFS. Secondly, the authors include a parameters that accounts for miss-specification of the ancestral allele. This is indeed crucial, since the excess of derived alleles is an important signal that the author try to infer, which would be heavily affected by the miss-specification.

The authors apply their method to simulated data, to demonstrate that it has power to infer the parameters of the model, and more importantly, correctly distinguish between Kingman's coalescent and multiple merger coalescents. The authors then compile around 45 SFS from different species across the tree of life. They infer the ancestral alleles based on respective outgroup populations, and also use standard approaches to scan for population structure, as this could also affect the SFS in ways that would not be captured by the vanilla Kingman coalescent. The authors can clearly reject population structure for roughly half of their datasets. They then apply their framework to infer the most likely genealogical model for each species, and detect a better support for a multiple merger coalescent in roughly 2/3 of the cases. The authors do discuss that this is not definitive proof of the sweepstake-like reproduction models that are used to motive the Psi- and Beta-coalescent, since there are other biological processes that can lead to multiple merger like genealogies, but they do conclude that in many cases, the genome-wide data is better described by a multiple merger coalescent.

The paper is a worthwhile contribution to understanding the genomic processes that shape genome-wide variation in different species. It demonstrates that in many cases, the standard Kingman coalescent is not the correct model to describe this variation, and future work will have to solidify what the exact mechanisms are in the different species that drive this deviation from the standard model. Compiling the different datasets and analyzing them in the newly introduced modelling framework is an impressive feat, and provides a good resource and reference for future studies. The statistical analysis is sound and convincing. I do, however, think that the manuscript can be improved to be more accessible to a general audience by more clearly introducing key concepts. In addition, I do think that there are some possible mechanisms that the authors did not address. While I do not think that this will invalidate the conclusions of the study, I do think it would strengthen the arguments. My apologies for the amount of text in this review. I did enjoy reading the article and just have a lot of suggestions.

Specifically:

- The excess of derived alleles is a key signal underlying the inference in this study. This signal would be severely affected, if the ancestral allele is miss-specified. The authors are aware of this and address it by using different outgroups for different species to polarize the segregating variants. In addition, they co-estimate a nuisance parameter for miss-specification probability, where they leverage tri-allelic sites (including outgroup) to refine the estimate. In the model that the authors fit, the miss-specification of ancestral alleles can only result from more than one mutation at the focal site when including the outgroup. I do not think that is the only mechanism that can lead to ancestral miss-specification. The other most important ones being sequencing error in the outgroup and incomplete-lineage-sorting.

Regarding sequencing error in the outgroup: If only a single sequence for the outgroup is used to polarize the SNPs, sequencing errors could readily affect the inference. In this case, a suitable model for sequencing error would be hard to incorporate, but one could include it in the framework as a single nuisance parameter for error probability. If there are multiple samples in the outgroup, then this problem is likely mitigated. Table A.5 contains a column "Diallelic outgroup," which seems to imply that at least for some outgroups, multiple samples are available, but it is not clearly stated in the manuscript whether this is the case. Please either include possible sequencing errors in the model or provide an argument (in the manuscript) why it does not affect the analysis.

Regarding incomplete-lineage-sorting or ancestral variation: If a certain SNP is already segregating in the ancestral population, then the T_MRCA is more ancient then the split of the focal population and the outgroup. At these SNPs, the infinite sites model in the focal population is invalid. It is unclear how these SNPs are treated in the analysis presented here. If a single sequence in the outgroup is chosen at such a SNP to polarize the focal sample, this can lead to miss-specification of the "ancestral" allele. Again, Table A.5 has a column titled "Diallelic outgroup," which I figure is related to the issue that I am describing. In some cases, there are sizeable amounts of SNPs in this category. However, it is unclear how or whether these SNPs are used in the analysis, or what the sample size of the outgroups is, which would allow the reader to see in how many cases only a single sequence is used in the outgroup. Please specify in the manuscript whether the "diallelic outgroup" are indeed SNPs (partially) resulting from incomplete-lineage-sorting. If so, state whether and how these SNPs are included in the analysis. Incomplete-lineage-sorting should either be included in the modeling framework, or a justification presented, why it is okay to not include it. Also, please provide sample sizes for the outgroup.

My apologies for the lengthy text and argument, which I hope was not entirely confusing. If I got something mixed up in my head, please tell me.

- In the article, the authors introduce the SFS, Kingman's coalescent, and report that this model is often a poor fit to observed U-shapes. The article then transitions to introducing the MMC models, and to the more technical details of the inference procedure. I think that a general audience would appreciate some heuristic explanations why MMCs can produce U-shapes before the inference framework is presented. I assume that there is some literature that exhibits these phenomena, so providing some references would be warranted. In addition, it would be good to insert a paragraph that explicitly states that MMCs can produce a U-shape, and perhaps some heuristical explanation, maybe even a figure or sketch comparing some theoretical spectra under MMCs and Kingman.

In the discussion and Appendix A.11, the authors present arguments, why Kingman's coalescent cannot produce the U-shape, even under arbitrary non-extreme demographic models. I think this arguments is a very good addition to the article. However, as a reader, I am now especially intrigued: Following the line of reasoning presented in A.11: What feature of MMCs leads to the U-shape? Do maybe all MMCs that don't have a Kingman component necessarily produce a U-shape? While a complete exposition is probably beyond the scope of the current manuscript, it would be good to provide some heuristics to the reader as to what mechanisms cause U-shapes, or provide references. It seems to me that MMCs have high (higher than Kingman) probabilities of unbalanced trees, and that can cause U-shapes. But I might be completely wrong.

- In their model selection framework, the author fit a single-parameter model of exponential growth (decline). While I do agree that this flexibility likely captures most of the relevant effects of the demographic history, I do think it would be good to add a few sentences in the discussion speculating on how the results of the study would be affected, if a more detailed demographic model would have been fit to the data.

- p.4, l.13: I find this population growth model a bit strange, but I appreciate that it is the most convenient way to introduce all parameters correctly. Could the authors add a sentence here to be explicit about the fact that under the "non-extreme" demography assumptions, the fact that one individual has U + G offspring does not lead to multiple mergers for U = 2, and does not change the distribution of the multiple mergers in the MMC models?

- In Section 2.2 and throughout the manuscript, the authors refer to the statistics for model comparison as "Bayes factors". I think this is a bit misleading, and should be changed. Bayes factors usually incorporate prior probabilities for the models being compared, and the authors never introduce such priors. The statistics they define are simply differences of log-likelihoods and thus most closely resemble likelihood-ratio statistics. One could argue that the statistics are Bayes factors with a uniform model prior, but I think this language should be avoided altogether. These statistics are comparing raw (composite) likelihoods, and the respective thresholds are shown to be well calibrated in the simulation study. Also, please add a sentence at the at of the second to last paragraph in Section 2.2, that the threshold log(-10) will be justified in the simulation study.

- In the first paragraph of Section 2.3, the authors state that "the collected SFS come from public data or private communications." This cannot be accepted for publication. All the datasets to re-create the study have to be accessible to the reader. The authors either have to provide the original data source, or provide unpublished datasets explicitly and include a description of how they were generated. Table A.5 provides citations for all but one dataset (Please explain what "RefSeq" means). Thus, it appears that most datasets do have an original source, and the "private communication" statement is not necessary? If the datasets were significantly altered from the original source, then the procedure should be provided explicitly. A more detailed description of the outgroup and citations for the outgroup are also needed, or a reference to the original publication of the dataset, if the outgroup is described there.

- Figure 1 (and most other plots showing estimation error): It is somewhat difficult to follow these plots, as the spacing of the grid causes gaps in the distribution of points. A possible improvement could be to indicate the grid density on the y-axis of each plot, so it is easily visually accessible. Also, an alternative to indicating the number of replicates with a certain value by the size of the circle, could be violinplots or histograms.

- Section 3.2, 3rd paragraph: Please provide a reference for Cramer's V and a reference for choosing the goodness-of-fit labels for certain ranges of the value. Please explain why it is necessary to account for different sample sizes across species. The value is computed for each species separately, so it seems that correcting for different sample sizes is not necessary.

- Figure 2: Please add a label (i \\xi_i) to the y-axis of the displayed spectra. It is specified in the caption, but it would be good to have it readily visible in the plots. Please replace the first sentence of the caption by "Estimates of \\alpha by species." It might be worth pointing out that, since the estimate of \\alpha for E. coli is 2, the uptick in the spectrum has to come exclusively from the allele miss-specification.

- Equation (4): I do think it is very helpful to include the argument of monotonic SFSs under any time-shifted Kingman coalescent. However, while the infinite sample size expression is helpful, I think it should be complemented with an argument for finite sample sizes (here or in A.11), since some sample sizes presented in the data analysis are rather small. There is no guarantee that, if a property holds for the limit, it also holds for each element of the converging sequence (not in general, that is).

Minor comments:

- p.2, l. 13: ... N the effective population size, ...

- p.2, l.-13: ... number of offspring per individual has variance on the order of the total population size.

- p.3, l.5: In this study, we collected SFS from 45 species...

- p.6, ;.2: We first demonstrate the power ..., and then apply it to ...

- p.6, l.7: ... for each parameter combination, choosing different values for the coalescent parameter, ...

- Table 1, caption, l. 4: The second column shows whether ...

- Table 2: What does MMC/Psi in the model column indicate?

- p.10, l.-6: ... we show that this approach has power to detect the correct MMC model ...

- p.10, l.-3: ... and orientation errors are additionally modeled, although in some cases ...

- p.11, l.-6 to -4: Non-extreme changes mean that the population size changes are on the same order of magnitude. [The current long-winded sentence in the manuscript is hard to follow.]

- p.11, l.-1: ... and a large sample is a linear function of the expected waiting times c_k for the next coalescence of k lineages, with a simple analytic form:

- p.22, l.-10: ... with distribution given in Section 2.1 leading to the ...

- p.24, l.-15: If one of the alleles is the same as the outgroup, ...

- p.24, l.-9: ... considerably longer branch lengths to the outgroup.

- p.24, l.-3: ... can be polarized (displaying one allele equal to the outgroup-allele) is 1-2p.

- p.25, l.-5: ... are independent of all other trees () ...

- Table 1 & Table A.2: Perhaps indicate the maximum in each row with boldface text, so it is readily visible.

- Figure A.6, Caption (a): Replace with "MMC parameter \\alpha or \\Psi", since the blackslash is used to indicate division.

Reviewer #3: The main objective of the article is to compare the empirically observed SFS from 45-genome, associated with different species across the tree of life, to the theoretical SFS expected. To be precise, they compare the undelying genealogy given by a coalescent with multiple merges (MMC), with the neutral null model. The MMC emerged to study populations where individuals can give birth to more than two offsprings. From the mathematical point of view, they are characterized by a measure on the unit interval. For the analysis the author focused in two particular classes: the Beta- and Psi- coalescent. Also the authors, assumed an exponential growth of the population and missorientation of the ancestral allele. To decide which coalescent model fit better, they computed the two log Bayes factors. They present different scenarios from which the MMC fits better compare with the Kingman and exihibts a U-shpaed SFS.

I recommed this paper for publication after a very minor revision because it is very well organized and explain their methodology clearly. More important their results are new and interesting. The Kingman coalescent has been used for awhile as a null model and the authors give a new framework where this is not very well applied. Also, the references are very helpful to see clearly their contributions and present an overview of the state of the matter.

Minor comments. In Table 2 it is not specified the meaning of

MMC/Psi. P. 3 L. 11 Explain the meaning of pure radiation. P. 9 L. -19 remove one "the" P. 24. L.-6 biallelic.

**Have all data underlying the figures and results presented in the manuscript been provided?**

Reviewer #1: Yes

Reviewer #2: **No: **It is unclear whether all datasets used are publicly available.

Reviewer #3: Yes

PLOS authors have the option to publish the peer review history of their article (what does this mean?). If published, this will include your full peer review and any attached files.

Reviewer #1: **Yes: **Yuseob Kim

Reviewer #2: No

Reviewer #3: No

---

## [Decision Letter · Decision Letter 1]

20 Dec 2022

Dear Dr Achaz,

Thank you very much for your care in the revisions to 'Interpreting the pervasive observation of U-shaped Site Frequency Spectra', submitted to PLOS Genetics.

The manuscript was fully evaluated at the editorial level and in independent peer review.  Both the associate editor and reviewer are satisfied with the revisions made, but have a few additional recommendations that should be addressed.  We therefore ask you to modify the manuscript according to the review recommendations. Your revision should, in particular, clarify Table A.7, as suggested by the reviewer.  The other minor points made by the reviewer should also be addressed.

Yours sincerely,

Lindi Wahl

Academic Editor

PLOS Genetics

Bret Payseur

Section Editor

PLOS Genetics

Reviewer's Responses to Questions

**Comments to the Authors:**

Reviewer #2: Review of "Interpreting the pervasive observation of U-shaped Site Frequency Spectra" by Fabian Freund, Elise Kerdoncuff, Sebastian Matuszewski, Marguerite Lapierre, Marcel Hildebrandt, Jeffrey D. Jensen, Luca Ferretti, Amaury Lambert, Timothy B. Sackton and Guillaume Achaz:

The authors have sufficiently addressed most of my comments. I think that the additional explanations why a U-shaped SFS is expected under MMCs helps non-specialist audiences to appreciate the findings. Moreover, the extensions of the model and additional analyses regarding other sources of miss-specification of the ancestral allele solidify the results and contribute to the compelling analysis.

I do, however, still feel that Table A.7 (previously A.5) needs to be presented better so that the description of the datasets and the analysis performed is clearer. In most rows, only one reference is given in the column "Source." Does this reference, in all cases, include a description of the outgroup? If not, a reference for the outgroup needs to be added. For M. tuberculosis, the "Source" column only indicates "RefSeq". Please provide a reference here, and if necessary, a date when the database was accessed.

Please add a clear description (perhaps in the caption) as to what the column "Diallelic outgroup" refers to. I assume it means number of SNP-sites in the ingroup where the outgroup shows two alleles. This a bit at odds with A. thaliana. Here, the outgroup size is 1, but there are 400,000 such sites. Is this a typo? Or is the outgroup 1 diploid individual that is heterozygous at all these sites?

Moreover, if there are two alleles in the outgroup, then there are 3 possible combinations with the ingroup:

1. Same two alleles in outgroup as in ingroup.

2. One allele shared between in- and outgroup, the other allele different.

3. No allele shared (probably very unlikely and perhaps not observed often).

Please replace the column "diallelic outgroup" by two columns that show the number for 1. and 2.

Moreover, for 2., it is clear what the inferred ancestral allele is. However, for 1., which is an ancestral polymorphism, what is the "ancestral" allele that is used in the analysis. How is it determined?

All of this might not be relevant, if I assume the wrong meaning of "Diallelic outgroup." But that would reinforce my point that it is necessary to define what is meant by it.

Minor comments:

- p.2, l.-5: ... recurring ancestor of a substantial fraction of the population ...

- p.2, l.-3: ... the presence of a single ancestor ...

- Section 3.2, third paragraph: Please insert an explicit reference to Appendix A.6 in this paragraph.

- p.12, l.-12: ... exact same patterns as natural mutations.

- p.23, l.6: Let us consider a reproduction model, for instance ...

- p.23, l.11: ... is the (random) tree obtained from tracing back the genealogy ...

- p.23, l.-14: Neutral genetic diversity results from neutral mutation events ...

- p.30, l.5: ... compare estimated diversity between in- and outgroups ...

- Supplement 1: The linestyles change between some of the plots. Please have all plots with unified linestyles.

- Supplement 1 & 2: I think it would be good to add an indicator in each plot as to what model was chosen as the most likely one.

**Have all data underlying the figures and results presented in the manuscript been provided?**

Reviewer #2: Yes

PLOS authors have the option to publish the peer review history of their article (what does this mean?). If published, this will include your full peer review and any attached files.

Reviewer #2: No

---

## [Editor Report · Decision Letter 2]

22 Feb 2023

Dear Dr Achaz,

We are pleased to inform you that your manuscript entitled "Interpreting the pervasive observation of U-shaped Site Frequency Spectra" has been editorially accepted for publication in PLOS Genetics. Congratulations!  Thanks for your care in completing the revisions; we hope you'll agree that the review process has allowed you to make an even stronger contribution to the journal and to our field.  We look forward to seeing this excellent paper in print.

Yours sincerely,

Lindi Wahl

Academic Editor

PLOS Genetics

Bret Payseur

Section Editor

PLOS Genetics

**Data Deposition**

http://datadryad.org/submit?journalID=pgenetics&manu=PGENETICS-D-22-00569R2

**Press Queries**

---

## [Editor Report · Acceptance letter]

19 Mar 2023

PGENETICS-D-22-00569R2 

Interpreting the pervasive observation of U-shaped Site Frequency Spectra 

Dear Dr Achaz, 

We are pleased to inform you that your manuscript entitled "Interpreting the pervasive observation of U-shaped Site Frequency Spectra" has been formally accepted for publication in PLOS Genetics! Your manuscript is now with our production department and you will be notified of the publication date in due course.

With kind regards,

Timea Kemeri-Szekernyes

PLOS Genetics

On behalf of:
